# Breaking the angular dispersion limit in thin film optics by ultra-strong light-matter coupling

Andreas Mischok [1,2] ✉, Bernhard Siegmund [3], Florian Le Roux[1], Sabina Hillebrandt [1,2], Koen Vandewal [3] & Malte C. Gather [1,2] ✉

Thin film interference is integral to modern photonics, e.g., allowing for precise design of high performance optical filters, photovoltaics and light-emitting devices. However, interference inevitably leads to a generally undesired change of spectral characteristics with angle. Here, we introduce a strategy to overcome this fundamental limit in optics by utilizing and tuning the exciton-polariton modes arising in ultra-strongly coupled microcavities. We demonstrate optical filters with narrow pass bands that shift by less than their half width (< 15 nm) even at extreme angles. By expanding this strategy to strong coupling with the photonic sidebands of dielectric multilayer stacks, we also obtain filters with high extinction ratios and up to 98% peak transmission. Finally, we apply this approach in flexible filters, organic photodiodes, and polarization-sensitive filtering. These results illustrate how strong coupling provides additional degrees of freedom in thin film optics that will enable exciting new applications in micro-optics, sensing, and biophotonics.

Two reflective surfaces form a basic Fabry-Pérot microcavity with a narrowband spectral response at a resonance wavelength $\lambda_{\mathrm{res},0}$ if the distance $d$ between the surfaces equals approximately $m\lambda_{\mathrm{res},0}/2n$, where $m$ is an integer and $n$ is the refractive index of the material between the cavity mirrors. If light impinges on this structure under an oblique angle $\theta$, the photon wavevector acquires an additional in-plane component, and thus the resonance shifts to a shorter wavelength, i.e., $\lambda_{\mathrm{res}}(\theta) = \lambda_{\mathrm{res},0}\sqrt{1-(\sin\theta/n)^2}$. This angular dispersion is considered to be a fundamental property of all thin optical films and must be taken into account during device design[1-8]. Thin film angular dispersion is sometimes utilized to fine-tune the spectral response of a device by tilting, but in practice thin-film angular dispersion is generally undesired for a number of reasons: (1) the presence of angular dispersion requires precise alignment and makes optical systems prone to drift over time; (2) in display applications, it can lead to changes in perceived color for different viewing angles; (3) if light with a distribution of angular components (as is the case for most light sources) passes through a thin film interference filter, the wavelength selectivity of the filter is compromised and the transmitted light is spectrally broadened in an uncontrolled way; (4) finally, the transmitted line-shape of optical components is often severely distorted at large angles, in part because the response of thin optical films depends on the polarization of the incoming light (polarization-splitting)[1,9].

So far, the main remedy for angular dispersion has been the use of high refractive index materials[10]. However, this approach is technically challenging, often introduces optical losses, and has limited efficacy. More elaborate designs for managing angular dispersion include dielectric and plasmonic nanostructures[11-17], lossy Fabry-Pérot cavities[16,18-23], and multilayer structures to compensate phase[24-26] or induce Fano resonances[27]. All these strategies however typically suffer from either high losses[28], low optical quality, or poor angular performance. Compared to conventional thin-film coatings, nanostructure-based filters also require a significant fabrication effort.

Here, we demonstrate how the fundamental angular dispersion limit in thin-film optics can be overcome by making use of the exciton-

---

[1]Humboldt Centre for Nano- and Biophotonics, Institute for Light and Matter, Department of Chemistry, University of Cologne, Greinstr. 4-6, Köln, Germany. [2]School of Physics and Astronomy, University of St Andrews, North Haugh, St Andrews, United Kingdom. [3]UHasselt, Institute for Materials Research (IMO-IMOMEC), Agoralaan, Diepenbeek, Belgium. ✉e-mail: andreas.mischok@uni-koeln.de; malte.gather@uni-koeln.de

polariton dispersion in strongly and ultra-strongly coupled microcavities[29–32]. Exciton-polaritons are quasiparticles that form through coherent interaction between an optical mode and an excitonic material resonance. The strongly coupled resonances split into a low energy (lower polariton) and high energy (upper polariton) branch once their coupling strength exceeds the losses of the individual resonances. The ultra-strong coupling regime is reached when this splitting becomes significant (~20%) relative to the bare exciton energy[30]; in the visible range, such splittings can equate to >100 nm in wavelength[33]. The characteristic shape of the polariton dispersion in optical microcavities has been studied extensively, and the distinctive anti-crossing between the lower and upper branch is often regarded as proof for the presence of strong coupling. It has been suggested that polariton dispersion can be exploited to improve color stability in information displays[33–35]; however, strong coupling has not been explored systematically for managing dispersion in thin-film optics. Strongly and ultra-strongly coupled exciton-polaritons have instead attracted great interest as a platform for studying light-matter interaction[32,36,37], with applications expected e.g., in polariton lasing[38–41], polariton chemistry[42], light emission[35,43–46], photodetection[47–49] and quantum information processing[50,51]. Organic semiconducting materials are of particular interest for these efforts as they can reach the ultra-strong coupling regime at room temperature[33] due to their exceptionally high oscillator strengths, broad absorption bands and large exciton binding energies. In addition, organic materials offer great spectral tunability and simple, low-cost and scalable fabrication.

We show here that by carefully adjusting the coupling strength and energetic offset (detuning) between an optical and an excitonic resonance, one obtains polaritonic dispersions that trace the angle-independence of organic excitons yet retain the spectral purity of an optical resonance. We demonstrate this concept by realizing narrowband transmission filters based on organic microcavities that operate in the strong and ultra-strong coupling regime. These filters exhibit spectral responses that are almost angle- and polarization-independent, with shifts in peak transmission wavelength over the full range of incident angles smaller than their respective linewidths. Utilizing different organic materials, we show the applicability of the concept throughout the visible spectral range and into the near infrared. As a further generalization, we introduce organic materials into multilayer dielectric stacks, which induces sideband coupling. This approach affords optical coatings that offer extinction ratios comparable to state-of-the-art dielectric filters, while severely suppressing the angle-dependence of their transmission spectra. Finally, to showcase the application potential of polariton-based filters, we explore three different scenarios: mechanically flexible filters with spectral characteristics that are robust to bending; integration in a sensitive, narrowband, angle-independent photodetector; and polarizing polariton filters utilizing anisotropic strong coupling.

## Results

### Ultra-strong coupling for dispersion management

Figure 1a schematically shows a transmissive narrowband Fabry-Pérot microcavity. If the material inside the cavity is transparent or weakly absorbing, the system is in the regime of weak light-matter coupling and the cavity exhibits a mostly parabolic dispersion with a substantial shift in resonance wavelength for different angles of incidence (Fig. 1b). Introducing a material with a strong excitonic resonance into the cavity gives rise to strong light-matter coupling and leads to a splitting of the original parabolic dispersion into two polariton branches: The lower polariton branch (LPB) is red-shifted with respect to both the photon and exciton resonances, shows approximately photonic parabolic dispersion at low angles, and flattens out to a more exciton-like dispersion at larger angles. The second, higher energy, upper polariton branch (UPB) shows the inverse behavior. Utilizing

strong coupling adds additional design parameters not available in conventional thin film optics design–namely the coupling strength and the cavity detuning. The coupling strength $\hbar\Omega_R/2$ describes the interaction between photons and excitons and thus the energetic (Rabi-)splitting ($\hbar\Omega_R$) between LPB and UPB. The detuning $\delta$ is the difference between the energies of the bare photon resonance at normal incidence ($E_P$) and bare exciton resonance ($E_X$). We find that tuning the microcavity to a relatively large absolute value of $\hbar\Omega_R$ and a moderate positive detuning $\delta$ in the range of 100 meV–200 meV results in a mostly flat LPB and thus largely avoid shifts in resonance wavelength with angle (Fig. 1d). These conditions also keep a high absolute transmission; even stronger positive detuning results in a loss of peak transmission while going to negative detuning makes the LPB more photonic and thus introduces strong angular dispersion again (Supplementary Information, Supplementary Fig. 13). Using a large $\hbar\Omega_R$ helps in two ways: first, it increases the separation between LPB and UPB, making it possible to filter out the UPB if desired (Supplementary Fig. 2). Second and more importantly, a large coupling strength (i.e., going into the ultrastrong coupling regime) creates a larger separation between the LPB and the bare exciton absorption, helping to avoid parasitic absorption from uncoupled excitons while keeping the overall dispersion flat (further details, see Supplementary Note 1 and 3).

Figure 1e illustrates this strategy by comparing transfer-matrix (TM) calculations of optical transmission versus angle and wavelength for different microcavities comprising a layer of either transparent $SiO_2$ or the organic absorber C545T sandwiched between two semi-transparent silver mirrors. The difference in the angle-dependent behavior is immediately obvious; the presence of ultra-strong coupling in the cavity with the organic absorber leads to a large splitting ($\hbar\Omega_R \geq 1eV$) and –in combination with the positive detuning– results in a near complete flattening of the dispersion compared to the uncoupled cavity based on $SiO_2$. (See Supplementary Table 2 for data on an uncoupled cavity based on high index $Ta_2O_5$).

Our concept is not limited to simple metallic cavities. Recently, it has been shown that excitonic resonances also couple with the Bragg sidebands of distributed Bragg reflector (DBR) cavities made of alternating high and low refractive index layers[52]. We utilize this behavior to design optical coatings with high extinction and sharp edge characteristics. TM simulations of a single DBR and a strongly coupled cavity containing a layer of C545T sandwiched by two similar DBRs illustrate how the Bragg sidebands of the mirror couple to the C545T exciton in a similar manner as the cavity mode in the previous example (Fig. 1f). However, in contrast to the metallic cavities, they now form multiple Bragg-polaritons[52,53], leading to a series of lower polariton branches, each with reduced angular dispersion.

To demonstrate the proposed polariton filter design experimentally, we choose several common organic semiconductors that show strong excitonic absorption throughout the visible spectrum, starting with the organic dye C545T already used in the above simulations (details in Supplementary Note 1). Figure 2a–f compares the measured, angle-resolved transmittance spectra of a conventional Ag-SiO$_2$-Ag microcavity filter to a Ag-C545T-Ag polariton filter. The C545T-based filter exhibits clear polariton formation with high transmission in the UPB and LPB, respectively (compare calculated polariton modes in Fig. 1e) and surpasses the conventional filter in all metrics: (1) it shows a sharper linewidth (27 nm vs 56 nm full width at half maximum (FWHM)), (2) a higher rejection on the blue side (minimum transmission 0.9% vs 5.7%), (3) a better spectral tolerance to thickness variations (Supplementary Fig. 5), (4) a drastically improved angular dispersion, and (5) higher transmission at large angles of incidence (Supplementary Fig. S11). While the conventional filter exhibits a spectral shift of >100 nm and a drastic increase in linewidth due to polarization splitting, the polariton filter shows a polarization-independent spectral shift of <15 nm and a stable linewidth even at

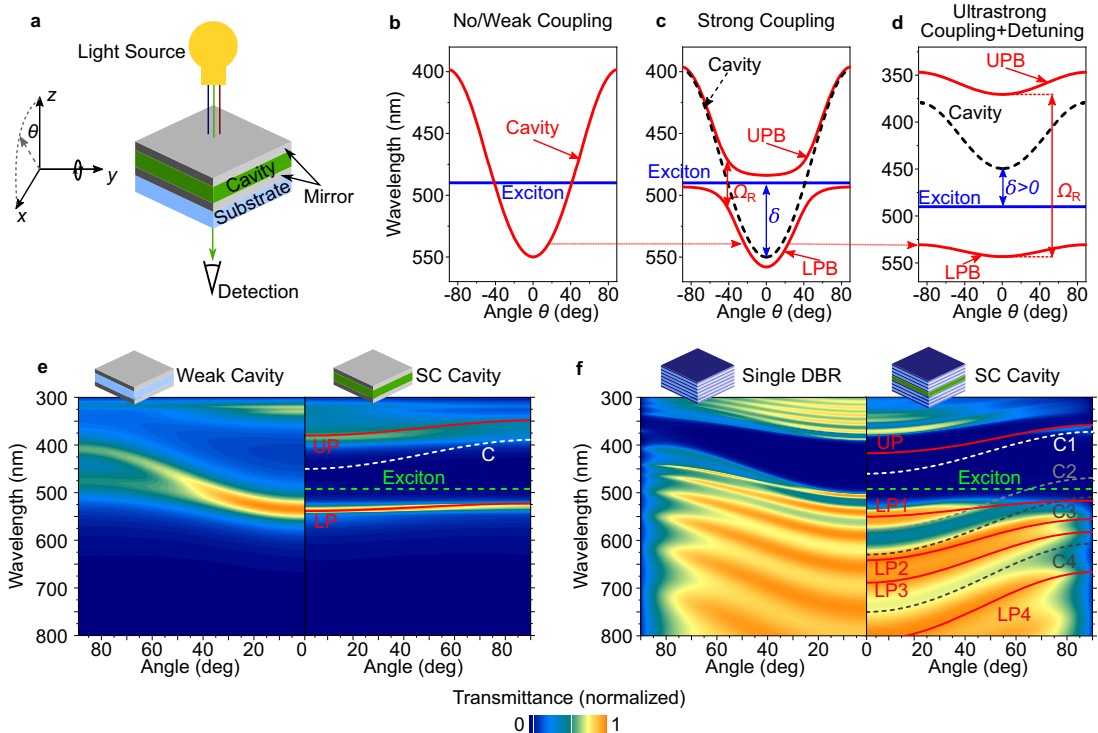

**Fig. 1 | General concept of polariton dispersion management and optical modeling of polariton filters. a** Sketch of cavity-based transmission filter between light source and detector, with a variable angle of incidence $\theta$. **b**–**d** Transfer matrix simulation of the angular dispersion of an uncoupled or weakly coupled cavity with intracavity refractive index $n \approx 1.45$ (**b**), a strongly coupled cavity with Rabi-splitting $\hbar\Omega_R \approx 0.2\,\mathrm{eV}$, negative detuning $\delta \approx -0.2\,\mathrm{eV}$ (**c**), and an ultra-strongly coupled cavity with $n \approx 1.85$, $\hbar\Omega_R \approx 1\,\mathrm{eV}$ and positive detuning of $\delta \approx 0.2\,\mathrm{eV}$ (**d**). Strong coupling leads to formation of upper and lower polariton branches (UPB/ LPB) which avoid the crossing point of cavity photon (dashed line) and material exciton (solid blue line). In each panel, the transmitted modes are indicated as solid red lines. While the uncoupled cavity shows a characteristic and large angular dependence, strong coupling can dramatically reduce this dependence; under ultra-strong coupling conditions in particular, the angular dispersion can be almost negligible. **e** Transfer matrix simulation of angle-resolved transmission of a metal-dielectric-metal weakly coupled (Weak) cavity (left) and a strongly coupled (SC) cavity (right). Angular dispersion leads to a significant blueshift and polarization splitting of the narrowband transmission for the weak cavity. For the SC cavity, the resonance is further narrowed and shows almost no blueshift and polarization splitting at large angles. Lines represent a coupled-oscillator model of polariton modes (red lines) as well as the bare exciton (green dashed) and cavity (white dashed) resonances. **f** Simulated transmission of a distributed Bragg reflector (DBR, left) and a blue-shifted strongly coupled DBR cavity (right). For the latter, the DBR sidebands couple to the excitonic resonance, leading to an angle-independent stopband between 400 nm and 500 nm.

extreme angles. For an angle range of 70°, its spectral shift is < 10 nm, while for the conventional filter, it is ≈100 nm. The enhanced angular stability is strikingly obvious when looking at the filter at different angles by eye (Fig. 2c, f). If transmission in the wavelength range of the UPB is undesired, it can be blocked by a complementary absorption layer (Supplementary Fig. 2).

While such basic cavity filters are easy to design, their performance and versatility, e.g., in terms of controlling lineshape, are limited. To tune the spectral response and enhance transmission, additional dielectric layers were added to the metal-metal polariton filters, with a final design of $Ta_2O_5(80\,nm)$-$SiO_2(62\,nm)$-$Ta_2O_5(58\,nm)$-$Ag(25\,nm)$-$C545T(100\,nm)$-$Ag(25\,nm)$-$SiO_2(176\,nm)$-$Ta_2O_5(71\,nm)$. This allowed us to shape the electric field in the cavity and thus reduce losses caused by parasitic absorption in the metal. Figure 2g–i showcases such a dielectric-supported filter that offers a broader passband (FWHM ≈ 85 nm), exhibits a peak transmission of 80% and shows a virtually unchanged transmission lineshape up to a 40° angle of incidence. In this design, there is a trade-off between peak transmission and linewidth, as the additional dielectric layers reduce the reflectivity of the metal layers and place them at a field minimum, which, in turn, slightly reduces cavity quality. This effect could be remedied by the addition of further layers to boost the overall reflectivity again.

Next, we performed a series of transfer matrix simulations to investigate if the absorption of the organic materials introduced to obtain strong light-matter coupling limits the peak transmission achievable in polariton filters. We find that for both conventional metal-metal microcavity filters and metal-metal polariton filters, the transmission performance is similar. For both architectures, performance is ultimately limited by residual absorption in the metallic layers and by reflection at the outer air-metal interface, with absorption in the organic materials not representing a significant loss pathway (absorption at peak < 3%, Supplementary Note 3 and Supplementary Figs. 11–16).

The chemical diversity of organic materials enables the application of the polariton filter concept across a wide spectral range. Figure 2j depicts the normalized extinction of the materials Spiro-TTB[54], BSBCz[55,56], C545T[57], Cl$_6$SubPc, and SubNc[58] (full chemical names in methods). We found that in general, materials with strong absorption and steep absorption onset are well suited to form exciton-polaritons with suitable properties to realize narrowband, angle-stable filters, while photoluminescence efficiency does not play a role. Therefore, different classes of organic materials, from fluorescent emitters (BSBCz, C545T) to photovoltaic absorbers (Cl$_6$SubPc, SubNc) and charge transport layers (Spiro-TTB), can be utilized to design such filters. As an example, we realized filters with transmission lines between 400 nm and 800 nm, with 35 nm thick Ag mirrors sandwiching the organic material (Fig. 2k). The transmission spectra of the different filters are well separated at all angles, as required for a

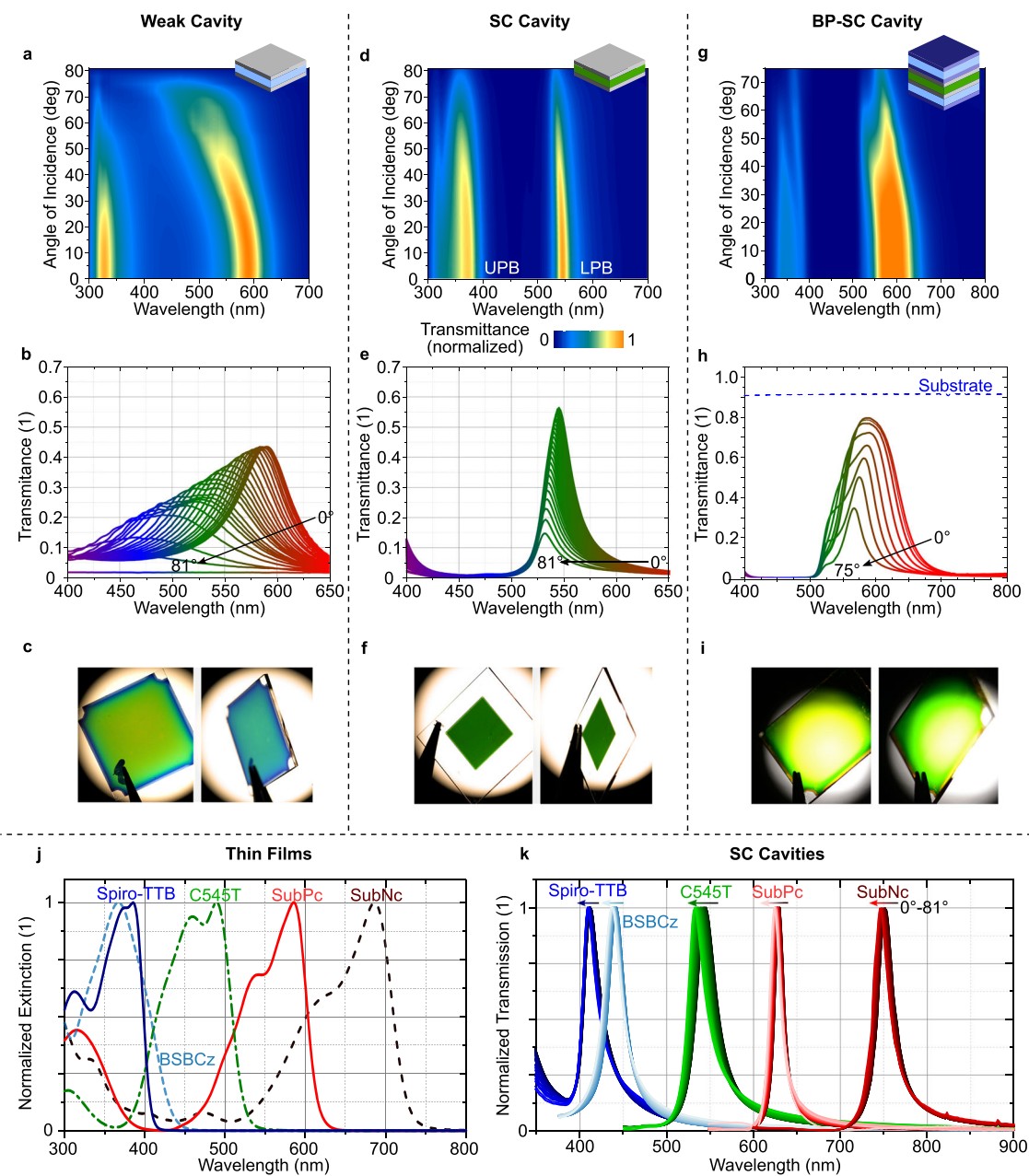

**Fig. 2 | Angle-resolved transmission of conventional and polariton-based metal cavity filters. a, b** Measured angle-resolved transmittance of a conventional, weak Ag(25 nm)-SiO₂(140 nm)-Ag(25 nm) cavity as false color map (**a**) and line plots (**b**). Due to dispersion, the position of the transmitted mode strongly depends on the angle of incident light, shifting by more than 100 nm and exhibiting strong polarization splitting at large angles. **c** Photographs of the weak cavity under normal and oblique angles. **d, e** Measured angle-resolved transmittance of an ultra-strongly coupled Ag(25 nm)-C545T(80 nm)-Ag(25 nm) cavity as false color map (**d**) and line plots (**e**). Ultra-strong coupling drastically reduces dispersion, resulting in a mode shift of < 15 nm over the entire measured angular range. **f** Photographs of the polariton filter under normal and oblique angles. **g, h** Measured angle-resolved transmittance of a bandpass filter (BP-SC) based on an ultra-strongly coupled Ag(25 nm)-C545T(100 nm)-Ag(25 nm) cavity with additional dielectric layers as

false color map (**g**) and line plots (**h**). Additional high (Ta₂O₅) and low (SiO₂) index dielectric layers are utilized to tune the transmission spectrum for a broader bandpass and higher peak transmittance (up to 80%), with high stability for angles up to 40°. Dashed line in (**h**) indicates transmission of the bare substrate. **i** Photographs of the bandpass polariton filter under normal and oblique angles. **j** Extinction spectra of five different organic materials used in this work. **k** Normalized transmission spectra of polariton filters based on the materials in (**j**). Each set of spectra shows the transmission of the LPB for angles of incidence from 0° to 81°, with an interval of 3°. By selecting suitable absorbers, polariton filters with narrowband angle-independent characteristics can be created across the whole visible spectrum, e.g., with Spiro-TTB (415 nm, blue), BSBCz (440 nm, light blue), C545T (540 nm, green), Cl₆SubPc (630 nm, light red) and SubNc (750 nm, dark red).

multitude of applications in spectroscopy, hyperspectral imaging, microscopy, and coloration. A detailed analysis of the metal-organic-metal filters, including full spectra, and variations in organic and metal layer thickness, is presented in Supplementary Note 1 and Supplementary Figs. 1–6.

## High-performance filters based on dielectric multilayers

As metal films introduce parasitic absorption, metal mirror-based filters are limited in terms of the maximum transmission and extinction they can achieve. Most applications requiring high optical quality, such as high passband transmission and high optical density in the

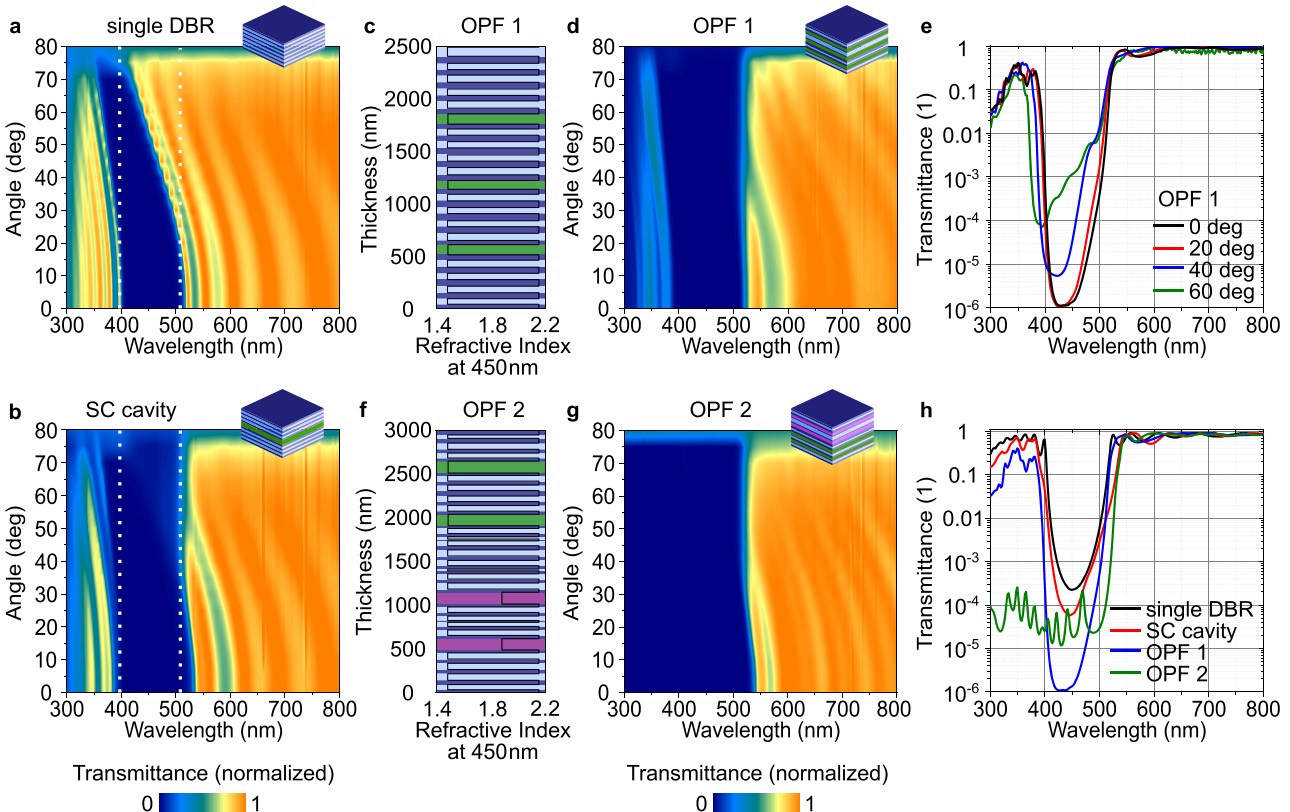

**Fig. 3 | DBR based polariton filters with high extinction. a–d** Experimental angle-resolved transmission spectra of a single DBR with 21 alternating layers of $Ta_2O_5$ and $SiO_2$ (**a**), and a strongly coupled cavity with two 11-layer DBRs sandwiching an organic C545T core layer (**b**). As a guide to the eye, dashed lines indicate the position of the stopband at 0°. **c** Layer stack of a computationally optimized filter design (Optimized Filter 1, OPF 1) comprising three layers of C545T (green), embedded in $Ta_2O_5$ (dark blue) and $SiO_2$ (light blue). **d** Experimental angle-resolved transmission spectra of OPF 1 in false color scale. **e** Transmission spectra of OPF 1 at different angles on a logarithmic scale. **f** Layer stack of a computationally optimized broadband filter design (Optimized Filter 2, OPF 2) comprising two Spiro-TTB layers (purple) and two C545T layers (green). **g** Experimental angle-resolved transmission spectra of OPF 2 on false color scale. While the stopband of the conventional DBR shows a strong angular dependence, the optimized polariton long-pass filters show largely angle-independent stopbands with high extinction and high transmission ( > 80%) at long wavelengths. **h** Transmission spectra of filters from (**a**–**g**) at normal incidence on a logarithmic scale.

stopband, therefore employ dielectric multilayer stacks, which comprise alternating films of low and high-refractive index materials. As already shown by the optical modeling above, embedding a material with a strong excitonic resonance in a cavity formed by dielectric mirrors can induce strong coupling with the sidebands of the mirrors. To further test this strategy, we fabricated and compared a single 21 layer DBR and a 2 × 11-layer DBR-based cavity with a core layer of C545T in the strong coupling regime (Fig. 3a, b). In contrast to previous demonstrations of strong coupling in DBR cavities, we strongly blue-shifted the design wavelength such that the red edge of the stopband coincides with the exciton resonance. The resulting side-band coupling flattens the DBR response at large angles and leads to a near-continuous stopband between 400 nm and 500 nm up to angles of incidence of 70°.

To further improve filter performance, we computationally optimized the individual layer thicknesses for angle-independent transmission using particle swarm optimization[59,60] followed by Levenberg-Marquardt optimization[61]. For Optimized Filter 1 (OPF 1, Fig. 3c, d), we introduced three 100 nm thick C545T layers to facilitate strong coupling and optimized the surrounding dielectric layers to achieve a high transmission >93% above 530 nm and a high rejection in the stopband, while limiting the total number of layers to 41, which is still low compared to many commercial filters. While fabrication-related uncertainties in the thicknesses of the individual layers led to some deviations from the predicted behavior, we were nevertheless able to experimentally demonstrate a filter with a largely angle-independent

stopband position that reaches an optical density (OD) of 6 and retains an OD of 4–5 even at large angles of incidence (Fig. 3e, compare simulations and conventional filter in Supplementary Fig. 7 and performance analysis in Supplementary Note 3). Furthermore, through the application of a backside anti-reflection-coating (ARC), the passband transmission can be increased to > 98% (Supplementary Fig. 15).

In a second optimization, we aimed to reduce the transmission of the blue sideband to create a longpass filter. To achieve this, we introduce a second organic material, Spiro-TTB, integrated into a blue-shifted DBR located on top of the C545T-based filter. Computational optimization led to a design comprising 2 × 2 organic cavities and a total of 55 layers. Experimental realization of this design yielded a longpass filter (OPF2) with a broad stopband OD > 3 across a large range of angles up to 70° (Fig. 3f, g). Figure 3h compares the measured transmission of the four presented filters on a logarithmic scale. (See Supplementary Note 2 and Supplementary Figs. 7–10 for further analysis.) These results illustrate how side-band coupling between highly absorbing materials and a dielectric multilayer stack can lead to a multitude of novel filter designs with drastically reduced angular dispersion. The performance of the DBR-based polariton filters discussed above is summarized in Table 1.

**Application scenarios of polariton filters**
The high angular stability offered by polariton filters enables their implementation as a mechanically flexible filter film[62,63], which is impractical for conventional interference filters as light is incident on

**Table 1 | Performance of DBR-based polariton filters**

| Filter-Type | Max $T$ | Max $T$ w/ ARC | Max OD | $\Delta\lambda$ at 45° | $\Delta\lambda$ at 75° | $\lambda_{edge}$ at 0° |
|---|---|---|---|---|---|---|
| | % | % | | nm | nm | nm |
| SC-Cavity | 92 | - | 4 | 21 | 15 | 530 |
| OPF1 | 93 | 98 | 6 | 4 | 19 | 525 |
| OPF2 | 93 | 98 | 4 | 19 | 16 | 531 |

the filter over a multitude of different angles when the filter is flexed, thus severely affecting its performance. To realize mechanical flexibility, we sandwich a polariton filter between two highly flexible, ultrathin quasi-substrates formed by Parylene-C and a protection and adhesion layer of $Al_2O_3$ that we obtain by an optimized chemical vapor deposition process[64] (Fig. 4a). The resulting filters have a total thickness of less than 15 µm and can be reversibly applied to many surfaces, including flat and curved optical elements. The angle-resolved transmission of flexible versions of the metallic polariton filter from Fig. 2d and the dielectric OPF2 design from Fig. 3f are shown in Fig. 4b, c, respectively. Like their counterparts on glass, they exhibit largely angle-independent spectral characteristics. Due to thin film interference in the thin quasi-substrates, their transmission spectra show additional high-order parabolic modes at longer wavelengths. While not an issue in applications like fluorescence microscopy, if needed, this effect could be reduced by tuning the thickness of the Parylene-C layers or moving to thicker substrates. We emphasize that even the complex design of OPF2, with 55 dielectric and organic layers, can readily be implemented in a flexible form factor, in part because the symmetric Parylene-C quasi-substrates place the filter in the neutral plane of the stack, thus reducing mechanical stress.

Next, we combine our polariton filters with broadband photodiodes. Instead of placing a filter at a macroscopic distance from the diode, we propose a monolithic design, where the diode is fabricated directly on top of the filter, with the light coming in through the substrate and the filter entering the diode over a range of angles (Fig. 4d). We use a C545T-based polariton filter and an organic photodiode (OPD) with 5 wt% TAPC diluted in $C_{70}$ as broadband absorber[65] (for diode characteristics see Supplementary Fig. 17). The entire structure is produced in a single vacuum deposition run and has a total thickness of 320 nm. Figure 4e compares the EQE of a reference diode without filter to the diode comprising the polariton filter. While the reference shows a broadband spectral response, the EQE of the polariton-filtered diode peaks at the positions of the LPB (450 nm) and UPB (380 nm) of the filter. Interestingly, due to the monolithic integration of filter and diode and the resulting optical coupling between them, the residual absorption of the polariton filter does not reduce the detection efficiency, and both diodes achieve comparable peak EQEs of ~18%. The coupling however also leads to a non-vanishing sensitivity between 400 nm and 500 nm, which is consistent with transfer matrix modeling of the standing electric field at the position of the $C_{70}$ layer (dashed red line in Fig. 4e). An additional, purely absorptive C545T can be integrated below the bottom mirror of the filter to avoid this effect (simulation, blue dashed line in Fig. 4e; a complete electric field simulation is presented in Supplementary Fig. 18). Finally, to demonstrate the advantage of our design over integration of a conventional interference filter, we perform an angle-resolved measurement of the EQE, where we observe an angle-independent spectral response for the monolithic device as expected (Fig. 4f).

Lastly, we use molecular orientation in films of a semiconducting polymer to induce anisotropic strong coupling[38,66] and, in turn create a polarization-dependent and angle-independent narrowband transmission filter (Fig. 4g). Poly(9, 9'-dioctylfluorene) (PFO) is aligned by depositing it on top of a Ag mirror coated with an alignment layer of Sulfuric Dye 1 (SD1). Subsequently, the resulting film is heated above

the liquid crystal transition temperature of PFO, followed by rapid quenching to room temperature to lock alignment[67]. Capping with a second Ag mirror completes the stack. As the transition dipole of PFO is mainly oriented along the polymer backbone, the strong coupling is only induced for light polarized parallel to the alignment of the polymer chains. This effect leads to a drastically different transmission response for the two possible polarizations. For light polarized perpendicular to the chain orientation, the sample is in the weak coupling regime and shows only a weak cavity mode at a wavelength of around 420 nm (Fig. 4h). By contrast, for light polarized parallel to the chain orientation, a narrowband and nearly angle-independent LPB appears at around 450 nm (Fig. 4i).

## Discussion

In summary, we demonstrated exciton-polariton-based optical filters with stable angular response using metal-organic-metal and multilayer dielectric cavities carefully designed to exploit ultra-strong coupling. These filters match or outperform comparable conventional filters in many performance metrics and show a blueshift smaller than their linewidth even at extreme angles of incidence > 80° (see Supplementary Fig. 16 and Supplementary Table 2, for a comparison to existing filters). Using a series of organic materials, we realized filters operating throughout the visible spectral range and into the near infrared, with even longer wavelengths likely also accessible in the future with appropriate absorber materials. Furthermore, we showed that ultrastrong coupling in dielectric resonators allows the design of high extinction optical filters, with an OD of up to 6 demonstrated here.

The amorphous nature of organic materials further allowed the fabrication of ultrathin filters that are mechanically flexible. In the future, such flexible filters might be mass-produced on carrier films to enable scalable and cost-efficient fabrication. These filters could then be applied to various non-planar optical components in a simple and reversible manner.

We also demonstrated monolithic integration of a polariton filter with a broadband organic photodiode to realize a thin-film photodetector (thickness < 500 nm) with non-dispersive and narrowband spectral response. We expect that integration with different polariton filters will allow to tailor the response of integrated photodetectors to the requirements of a wide range of applications. Beyond the integration with organic photodetectors, polariton filters might also be deposited or laminated onto charge-coupled-device (CCD) and complementary metal-oxide semiconductor (CMOS) cameras, thus enabling the use of high numerical aperture optics in spectral detection and hyperspectral imaging[68].

Finally, we utilized anisotropic strong coupling to design polarizing polariton filters. Like for the non-polarizing filter designs above, using films of aligned materials with different absorption profiles[66] will allow to produce polarization-dependent polariton filters with tailored spectral characteristics.

The use of ultra-strong light-matter coupling to manipulate angular dispersion differs fundamentally from a system where an angle-dependent thin-film interference filter and angle-independent absorption filter are stacked on top of each other, in the sense that strong light-matter interaction redistributes the spectral bands defined by the interference rather than just adding absorption in a specific spectral region (see Supplementary Fig. 12 for a direct comparison of the two approaches). Among other advantages, this allows a reduction of the amount of absorptive material used and retains the design freedom offered by dielectric filters. In addition, polariton filters also show considerably improved stability against inhomogeneities and variations in film thickness and, due to their angle-independence, are more robust to misalignment than conventional dielectric filters.

The strong absorption of organic materials allows for the creation of ultrathin polariton filters (metal-organic-metal < 200 nm, DBR < 4 µm) and the breadth of organic materials facilitates the

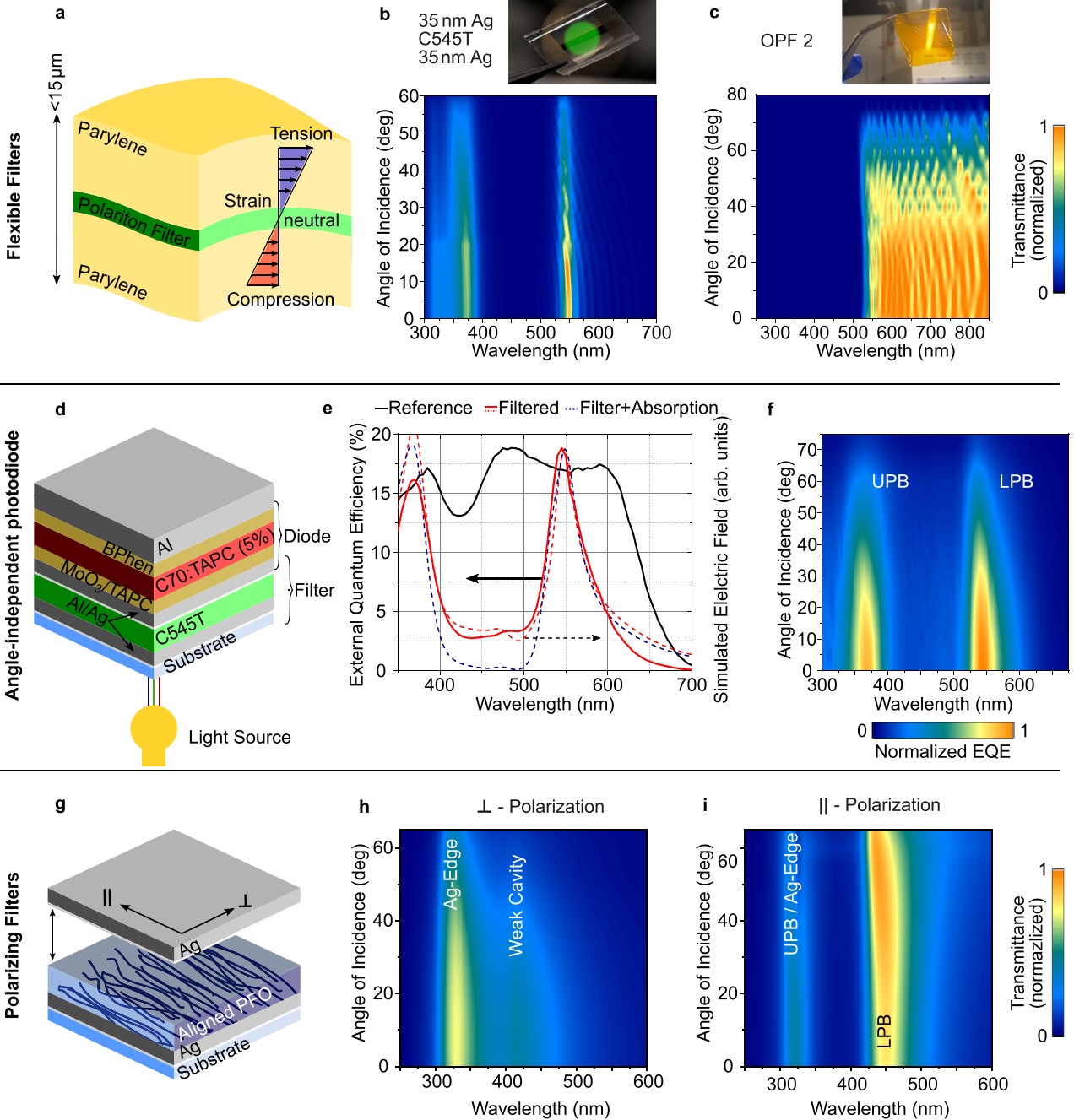

**Fig. 4 | Variations of polariton filters. a** Illustration of an ultrathin flexible filter design comprising two 6 μm thick parylene quasi-substrates and a C545T-based polariton filter. **b** Photograph and measured angle-resolved transmission of flexible metal-organic-metal polariton filter. **c** Photograph and measured angle-resolved transmission of a flexible variant of Optimized Filter 2 (OPF2). **d** Device structure of monolithically integrated photodiode and polariton-filter. **e** Experimental external quantum efficiency spectra (solid lines) of a reference photodiode (black) and the photodiode with integrated polariton filter (red) and corresponding simulated optical field amplitudes within the C70:TAPC heterojunction (dashed lines). The remaining response between 400 nm and 500 nm can be reduced by adding an absorbing C545T layer below the filter stack (simulation, dashed blue line). **f** Measured, angle-resolved external quantum efficiency of photodiode with integrated polariton filter, showing the angle-independent narrowband response at the positions of the UPB and LPB. **g** Illustration of polarizing polariton filter with strong coupling mediated by an aligned layer of PFO polymer. **h, i** Filter transmission for polarization perpendicular (**h**) and parallel (**i**) to the polymer chain orientation. Due to the polarization-dependent absorption of aligned PFO, strong coupling to the cavity mode only occurs for light polarized parallel to the polymer chains, leading to a polarization-dependent LPB and, thus, a highly polarization selective filter.

design of polariton filters with a wide range of spectral response. In addition, the concept is likely not limited to organic materials; strongly interacting perovskites, quantum dots and other low-dimensional systems, as well as inorganic structures can all be explored for different wavelength regimes and to add additional functionality.

In the future, our filter concept might be enhanced further, e.g., by introducing strongly absorbing thin films into more complex multi-layer coatings with designs tailored to the needs of specific high-performance optics. Thin-film interference filters with minimal levels of angular dispersion have long been sought after by optical engineers and manufacturers of optical coatings, but thus far they can only be

realized to a very limited degree by using expensive, high-index materials and high-precision deposition systems or by very extensive patterning. Our approach thus opens a new parameter space for optical coatings and the design of optical systems in general. We envision that angle-independent polariton filters will be of particular relevance to micro-optics, sensing[69], display applications[35], and biophotonics[70]. In all these areas, it is often impossible or impractical to work with collimated beams of light and, therefore, light is frequently incident on filter elements over an extended range of angles. Polariton-based optical filters will likely disrupt the design rules for such systems and allow for ground-up redesigns with improved performance as well as reduced size and complexity.

## Methods

*Sample fabrication:* Metal-organic-metal polariton filters were fabricated via thermal evaporation of organic and metal thin films at a base pressure of $1 \times 10^{-7}$ mbar (Angstrom EvoVac) onto 1.1 mm-thick glass substrates. The materials used were Al and Ag as metallic reflectors (Kurt J. Lesker Co.), 2, 2′, 7, 7′-tetrakis(N,N′-di-p-methylphenylamino) −9, 9′-spirobifluorene (Spiro-TTB), 4, 4′-Bis(4-(9H-carbazol-9-yl)styryl) biphenyl (BSBCz), 2, 3, 6, 7-tetrahydro-1, 1, 7, 7,-tetramethyl-1H, 5H, 11H-10-(2-benzothiazolyl)quinolizino-[9, 9a,1gh]-coumarin (C545T), 2, 3, 9, 10, 16, 17-hexachlorinated boron subphthalocyanine chloride ($Cl_6SubPc$) and Boron sub-2, 3-naphthalocyanine chloride (SubNc) as strong coupling layers and $MoO_3$, 1,1-Bis[(di-4-tolylamino)phenyl] cyclohexane (TAPC), Fullerene C70, 4, 7-Diphenyl-1,10-phenanthroline (BPhen) for photodiode fabrication. All organic materials and $MoO_3$ were obtained from Lumtec in sublimed grade and used as received. For dielectric polariton filters, $Ta_2O_5$ and $SiO_2$ films were produced by radiofrequency magnetron sputtering from a $Ta_2O_5$ or $SiO_2$ target (Angstrom) at a base pressure of $1 \times 10^{-7}$ mbar, using 18 standard cubic centimeters per minute (sccm) Argon flow at 2 mTorr process pressure. For $Ta_2O_5$, an additional oxygen flow of 4 sccm was added to prevent the formation of suboxides. Film thicknesses were controlled in situ using calibrated quartz crystal microbalances (QCMs). Before fabrication, substrates were cleaned by ultrasonication in acetone, isopropyl alcohol and deionized water (10 min each), followed by $O_2$ plasma-ashing for 3 min. Flexible filters were prepared on cleaned glass carrier substrates, similar to the process described in Keum et al[64]. In brief, parylene-C (diX C, KISCO) and $Al_2O_3$ layers were deposited using a parylene coater (Parylene P6, Diener electronic) and an atomic layer deposition (ALD) reactor (Savannah S200, Veeco) which are connected to the evaporation chamber via a nitrogen filled glovebox. After processing, the devices were peeled off the carrier substrate to yield free-standing, flexible filters. Filters were used in air without further encapsulation. Photodiodes were encapsulated in a nitrogen atmosphere with a glass lid using UV-curable epoxy (Norland NOA68), preventing intermittent exposure to ambient air. The active area of these devices was 16.0 mm². For polarizing filters, a layer of sulfuric dye 1 (SD1, Dai-Nippon Ink and Chemicals, Japan)[38] was spin-coated, annealed for 10 min at 150 °C, and exposed to 5 mW of polarized UV light for 10 min to photo-align the film. Films of poly(9, 9-dioctyl-fluorene) (PFO, Sumitomo Chemical Company) were spin-coated on top of the SD1 film in an inert environment, using 8 mg ml$^{-1}$ PFO in toluene solution. The sample was placed on a precision hotplate in an inert environment and the temperature was raised from 25 to 160 °C at a rate of approximately 30 °C min$^{-1}$. The upper temperature was then held for 10 min, followed by rapid quenching to room temperature by placing the sample on a metallic surface.

*Computational design of multilayer stacks:* Positioning of the organic layers inside the multilayer stack was first optimized using a particle swarm optimization algorithm[59] combined with the needle method[71]. This was followed by a further optimization using the Levenberg-Marquardt algorithm implemented in OpenFilters[61]. The number and thickness of layers were restricted to ensure an experimentally viable design. When fabricating the optimized designs, a thickness uncertainty of ~5% is introduced due to fabrication imperfections related to QCM tooling and the complexity of the designs. Industrial optical coatings routinely achieve better thickness control than what we had access to for this work.

*Device characterization:* Optical constants of all used materials and angle-resolved transmission measurements of the filters were obtained with a J.A. Woollam M2000 varying angle spectroscopic ellipsometer. Measurements at high optical density were performed using a two-beam spectrophotometer (Lambda 1050 + , Perkin-Elmer). For verification, measurements were repeated three times; there were no observable differences between measurements within the same sample. All transmission values shown represent external transmission, i.e., the total fraction of light making it through the entire filter, including its substrate. Photodiodes were evaluated using a xenon-lamp, combined with a monochromator (Oriel Cornerstone 130 1/8 m), a chopper and a lock-in amplifier (SR830 DSP, Stanford Research Systems). The excitation intensity was monitored by a calibrated silicon reference diode. Device structures and electric field distributions of the photodiodes were simulated using a transfer matrix model. Polariton branches were calculated using a coupled oscillator model[57]. Photographs were taken using a digital single-lens reflex camera (Nikon D7100) with a macro lens (Sigma 105 mm F2.8 EX DG OS HSM).

## Data availability

The data generated in this study are openly available via the St Andrews Research Portal at https://doi.org/10.17630/08c811a6-e77b-4ea8-aac2-86a808006b43[72].

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

## Acknowledgements

We are grateful to Prof. Klaus Meerholz for providing access to the variable angle spectroscopic ellipsometry setup and to Prof. Donal Bradley and the Sumitomo Chemical Company for provision of PFO. This research was financially supported by the Alexander von Humboldt Foundation (Humboldt Professorship to M.C.G.) and the European Research Council under the European Union's Horizon Europe Framework Program/ERC Advanced Grant agreement no. 101097878 (HyAngle, to M.C.G.) and the ERC grant agreement no. 864625 (ConTROL, to K.V.). A.M. acknowledges funding through an individual fellowship of the Deutsche Forschungsgemeinschaft (no. 404587082, to A.M.), from the European Union's Horizon 2020 research and innovation program under the Marie Skłodowska-Curie grant agreement no. 101023743 (PolDev, to A.M.) and from the Bundesministerium für Bildung und Forschung (BMBF) within a GO-BIO initial project no. 16LW0454 (FluoPolar, to A.M.). B.S. acknowledges funding from the Research Foundation – Flanders (FWO) through a Senior Postdoctoral Fellowship no. 12AOC24N (SOFIA, to B.S.). F.L.R. acknowledges funding from the Alexander von Humboldt Foundation through a Humboldt Fellowship.

## Author contributions

A.M. and M.C.G. designed the study. A.M. and S.H. fabricated all devices. A.M. measured and analyzed ellipsometry and angle-resolved transmission data. A.M., B.S and K.V. designed photodetector devices, which were measured by B.S. A.M. and F.L.R. designed and fabricated polarizing filters. All authors evaluated and discussed the results. A.M. and M.C.G. wrote the manuscript with input from all authors.

## Funding

## Competing interests

A.M. and M.C.G. are inventors of two pending patent applications (application numbers DE102020125597 and DE102024109482.6) filed by the University of Cologne on polariton-based filters. The remaining authors declare no competing interests.
