## [Transparent Peer Review file · Nature Communications]

Breaking the angular dispersion limit in thin film optics by ultra-strong light-matter coupling

Corresponding Author: Professor Malte Gather

Version 0:

Reviewer comments:

Reviewer #2

(Remarks to the Author)

I have carefully read through the revised manuscript and the authors' point-by-point response to all the comments. Overall, this research demonstrated some improvement of the angular dispersion of FP filters. This is essential to publish a work, but it does not guarantee publication on a high impact journal. I consider that this work may be more suited for a more specialist journal due to the following concerns on both working mechanism and filter performance.

--Comment on the mechanism.

I suspect that the claimed angular insensitive property of this filter may simply be caused by the absorption of the organic materials rather than the exciton-polariton modes in ultra-strongly coupled microcavities as discussed in the main text. Looking carefully at Figure 2j and 2k, the absorption band edge of each organic material (roughly 400 nm for Spiro-TTB, 500nm for C545T, 600nm for SubPc, 700nm for SubNC) matches well with the short wavelength edge of the passband of each associated filter. With the increase of the incident angle, the resonance of the FP cavity shows a blue shift. Light at the wavelength in the organic material's absorption band is absorbed, which limits the blue shift of the resonance, and thus the peak transmission of the passband quickly reduces with the increasing angle as shown in Figure 2e. If the above assumption is correct, the absorptive organic material can be put outside the conventional FP cavity. In this case, incident light goes through the FP first and then is filtered by the organic material to remove the short-wave components. I suggest the authors to do such a simple calculation and compare the results to the proposed SC cavity. The absorption of organic material in cavity may be higher than that placed outside due to the cavity enhancement, which may be considered in the calculation.

--Comment on the performance

Low transmission at a large incident angle weakens the significance of this research. As a major concern of all reviewers, the transmission drops quickly with the increasing incident angle. In Figure 2e, increasing incident angle from 0 deg to 81 deg, the peak transmission of SC cavity drops from 57% to 15%, and that of weak cavity drops from 43% to 10%. The decay rate is similar. The authors presented the results of a band-stop filter in Figure 3 to address this concern. I cannot accept it. It is easy to achieve high transmission at a wavelength outside the stop band in a band-stop filter. The reviewers' concern is about the bandpass filters shown in Figure 2. If there are strategies to improve the performance, it should be done on an example of a bandpass filter. In fact, for a band-stop filter, what we really care about is transmission inside the stop band, which increases quickly with the increasing angle in Figure 3e. So, transmission in the stop/pass band changes quickly with the incident angle in a band-stop/bandpass filter. They share the similar issue.

Reviewer #3

(Remarks to the Author)

The manuscript has been extensively revised and improved over the rounds of revisions. It will make a great contribution for Nature Communications and I recommend publication of the paper without further changes.

Reviewer #4

(Remarks to the Author)

Having reviewed the previous version of this manuscript, I believe it is high quality work and that it is a good fit for publication in Nature Communications.

Response Letter: Breaking the angular dispersion limit in thin film optics by ultra-strong light-matter coupling (ms NCOMMS-24-64048-T)

Reviewer #2 (Remarks to the Author):

I have carefully read though the revised manuscript and the authors' point-by-point response to all the comments. Overall, this research demonstrated some improvement of the angular dispersion of FP filters. This is essential to publish a work, but it does not guarantee publication on a high impact journal. I consider that this work may be more suited for a more specialist journal due to the following concerns on both working mechanism and filter performance.

Reply: We thank the reviewer for their careful reading of the manuscript. However, we still disagree with the assessment that the work is only of interest for specialised audiences. We have addressed similar concerns in previous rounds of revision. We hope to address the reviewer's remaining comments in the following:

Comment #1: Comment on the mechanism.

I suspect that the claimed angular insensitive property of this filter may simply be caused by the absorption of the organic materials rather than the exciton-polariton modes in ultra-strongly coupled microcavities as discussed in the main text.

Looking carefully at Figure 2j and 2k, the absorption band edge of each organic material (roughly 400 nm for Spiro-TTB, 500nm for C545T, 600nm for SubPc, 700nm for SubNC) matches well with the short wavelength edge of the passband of each associated filter. With the increase of the incident angle, the resonance of the FP cavity shows a blue shift. Light at the wavelength in the organic material's absorption band is absorbed, which limits the blue shift of the passband. It therefore seems that the structure has a small angular dispersion. Actually, it does not restrain the blue shift of the resonance, and thus the peak transmission of the passband quickly reduces with the increasing angle as shown in Figure 2e.

If the above assumption is correct, the absorptive organic material can be put outside the conventional FP cavity. In this case, incident light goes through the FP first and then is filtered by the organic material to remove the short-wave components. I suggest the authors to do such a simple calculation and compare the results to the proposed SC cavity. The absorption of organic material in cavity may be higher than that placed outside due to the cavity enhancement, which may be considered in the calculation.

Reply: It is true that the organic material used for polariton filters is selected to have its absorption edge close to the desired spectral position of the lower polariton band (LPB) and hence the passband or edge of the filter. (This design rule is already discussed in the manuscript.) However, the observed behaviour in our polariton filters, which are ultra-strongly coupled micro-cavities, is fundamentally different from a simple combination of an FP cavity (referred to as MDM structure in our manuscript) and an absorptive filter. For the latter, the blue-shifted transmission at higher angles would simply be absorbed, resulting in a sharp drop in transmission without a remaining detectable mode. Instead, in polariton filters, light is redirected into the polariton branches. In this way, the LPB retains a high and spectrally stable

transmission over a much larger angle range, while transmission in the blocking region between LPB and UPB is still strongly suppressed.

To demonstrate the difference between our polariton filter and a combination of an MDM structure with an organic material placed outside that MDM cavity, we have added Figure S12 to the revised SI (also reproduced below), which shows the angle-resolved transmission of a SubPc-based polariton filter (a), a conventional MDM filter with SiO₂ as the dielectric (b), a bare SubPc layer used as a purely absorptive filter (c), as well as the combination of MDM and organic absorptive filter located outside the MDM structure; for the latter, we distinguish the case where the organic material is outside but in direct contact with the MDM cavity (d) and the case where the organic material and the MDM cavity are on opposite sides of the substrate (e). As is clear from the figure, the polariton filter shows a strongly improved angular performance, allowing for transmission of light at a stable wavelength even for high angles, while all other filter variants fail (also see Fig. S12f which compares 0° and 45° transmission of the different variants).

To address the last point of the reviewer: As the transmission of a bare 80 nm SubPc layer is already <5 % in the absorptive region (Fig. S12 c), an increased absorption of the organic layer when placed inside the cavity cannot be the reason for the observed behaviour. In addition, we find that further increasing the SubPc layer thickness does not lead to better performance for the combination of MDM cavity and organic absorptive filter located outside it.

Figure S12 Comparing polariton filters against MDM filters, absorptive filters and combined MDM / absorptive filters. a-e Angle-resolved transmission of a 25 nm Ag | SubPc | 25nm Ag polariton filter, b 25 nm Ag | SiO₂ | 25nm Ag MDM-filter, c 80 nm SubPc absorptive Filter, d 80nm SubPc | 25 nm Ag | SiO₂ | 25nm Ag combined absorptive MDM filter with both elements directly next to each other allowing for some degree of optical coupling between the layers, e 80nm SubPc | Substrate | 25 nm Ag | SubPc | 25nm Ag combined absorptive/MDM

filter without optical coupling between the two elements. **f** Direct comparison of SubPc polariton filter (red), SiO₂-MDM (grey) and coupled SiO₂-MDM + SubPc (blue) at 0° (solid lines) and 45° (dashed lines) angle of incidence. The polariton filter acts fundamentally different to a combination of a conventional dielectric and absorptive filters, as the transmission is redirected towards the polariton branches instead of just being passively absorbed, resulting in a high and spectrally stable transmission at all angles.

Changes: We have added this discussion and the above figure to the revised manuscript.

Comment #2: --Comment on the performance

Low transmission at a large incident angle weakens the significance of this research. As a major concern of all reviewers, the transmission drops quickly with the increasing incident angle. In Figure 2e, increasing incident angle from 0 deg to 81 deg, the peak transmission of SC cavity drops from 57% to 15%, and that of weak cavity drops from 43% to 10%. The decay rate is similar. The authors presented the results of a band-stop filter in Figure 3 to address this concern. I cannot accept it. It is easy to achieve high transmission at a wavelength outside the stop band in a band-stop filter. The reviewers' concern is about the bandpass filters shown in Figure 2. If there are strategies to improve the performance, it should be done on an example of a bandpass filter. In fact, for a band-stop filter, what we really care about is transmission inside the stop band, which increases quickly with the increasing angle in Figure 3e. So, transmission in the stop/pass band changes quickly with the incident angle in a band-stop/bandpass filter. They share the similar issue.

Reply: We would like to draw the reviewer's attention to Figure S11 and the related Supplementary Discussion 3 in the SI. Here, we addressed the comparison between these filter types in detail; we find that the LPB mode is not only a spectrally stable but also that it remains highly transmissive at large angles.

To briefly sum up previous discussions related to this point: Any reduction in transmission at high angles of incidence is in large part due to the quality of the reflector (here the silver layer). Unfortunately, our experimental system is limited in the quality of the vacuum produced and evaporation rate, leading to a poorer quality silver film than is achieved for high quality silver layers produced in the optical coating industry. In turn, the transmission at large angles reduces due to the increased interaction with the metal film. However, as the reviewer notes, the loss in peak transmission at large angles is comparable to or better than the conventional MDM filter, meaning no additional loss is introduced while the spectral stability is dramatically improved. To further address the question regarding performance at high angle, in Fig. S11 we show simulations of high-quality (but experimentally achievable!) metal-metal cavities, showing that the polariton filter can not only compete but clearly outperform conventional filters in absolute transmission at high-angle, showing a peak transmission of >60% over the full angle range.

Reviewer #3 (Remarks to the Author):

The manuscript has been extensively revised and improved over the rounds of revisions. It will make a great contribution for Nature Communications and I recommend publication of the paper without further changes.

Reply: We thank the reviewer for their assessment and positive evaluation of the manuscript.

Reviewer #4 (Remarks to the Author):

Having reviewed the previous version of this manuscript, I believe it is high quality work and that it is a good fit for publication in Nature Communications.

Reply: We thank the reviewer for their assessment and positive evaluation of the manuscript.